# A Scoping Review on Body Fluid Biomarkers for Prognosis and Disease Activity in Patients with Multiple Sclerosis

**DOI:** 10.3390/jpm12091430

**Published:** 2022-08-31

**Authors:** Nadia Barizzone, Maurizio Leone, Alessandro Pizzino, Ingrid Kockum, Filippo Martinelli-Boneschi, Sandra D’Alfonso

**Affiliations:** 1Department of Health Sciences, UPO, University of Eastern Piedmont, 28100 Novara, Italy; 2Center for Translational Research on Autoimmune and Allergic Disease (CAAD), UPO, University of Eastern Piedmont, 28100 Novara, Italy; 3Neurology Unit, Fondazione IRCCS Casa Sollievo Della Sofferenza, 71013 San Giovanni Rotondo, Italy; 4Neuroimmunology Unit, Department of Clinical Neuroscience, Center for Molecular Medicine, Karolinska Institute, 17176 Stockholm, Sweden; 5IRCCS Fondazione Ca’ Granda Ospedale Maggiore Policlinico, Neurology Unit and Multiple Sclerosis Centre, Via Francesco Sforza 35, 20122 Milan, Italy; 6Dino Ferrari Center, Department of Pathophysiology and Transplantation, University of Milan, Via Francesco Sforza 35, 20122 Milan, Italy

**Keywords:** multiple sclerosis, biomarkers, prognosis, clinically isolated syndrome, body fluid

## Abstract

Multiple sclerosis (MS) is a complex demyelinating disease of the central nervous system, presenting with different clinical forms, including clinically isolated syndrome (CIS), which is a first clinical episode suggestive of demyelination. Several molecules have been proposed as prognostic biomarkers in MS. We aimed to perform a scoping review of the potential use of prognostic biomarkers in MS clinical practice. We searched MEDLINE up to 25 November 2021 for review articles assessing body fluid biomarkers for prognostic purposes, including any type of biomarkers, cell types and tissues. Original articles were obtained to confirm and detail the data reported by the review authors. We evaluated the reliability of the biomarkers based on the sample size used by various studies. Fifty-two review articles were included. We identified 110 molecules proposed as prognostic biomarkers. Only six studies had an adequate sample size to explore the risk of conversion from CIS to MS. These confirm the role of oligoclonal bands, immunoglobulin free light chain and chitinase CHI3L1 in CSF and of serum vitamin D in the prediction of conversion from CIS to clinically definite MS. Other prognostic markers are not yet explored in adequately powered samples. Serum and CSF levels of neurofilaments represent a promising biomarker.

## 1. Introduction

Multiple Sclerosis (MS) is an autoimmune demyelinating disease of the central nervous system (CNS) with a complex etiology. It is an invalidating disease that highly impacts the quality of life of patients and of public health systems with direct and indirect costs. It has a chronic course that evolves over 30 to 40 years and a variable phenotype spectrum, characterized for 85–90% of patients by the acute onset of transient neurological symptoms (relapsing-remitting course, RRMS), that in the majority of cases evolves into a progressive disability with or without superimposed relapses after 10–15 years from onset (secondary progressive form, SPMS). In 10–15% of patients, instead, the disability progressively deteriorates from the disease onset without acute episodes (primary progressive MS, PPMS). The therapeutic options for SPMS patients are limited when compared to those available for RRMS patients, therefore it is important to provide an early prediction of clinical course to decide on whether to start a first-line or second-line aggressive drug option from disease onset (Available online: https://www.nationalmssociety.org/ (accessed on 19 August 2022)).

Clinically isolated syndrome (CIS) refers to the first clinical episode suggestive of demyelination of the CNS, while radiologically isolated syndrome (RIS) characterizes individuals who present with incidental brain MRI findings similar to those observed in patients with MS, but who clinically have no signs of MS. Patients with CIS or RIS are at high risk (around 59% over 4 years for CIS and 28% over 5 years for RIS) of developing either RRMS or primary progressive (PPMS) MS [1,2].

Several biomarkers are increasingly suggested in MS as having potential use in clinical practice. They include body fluid biomarkers, neuroimaging biomarkers and even clinical biomarkers with different clinimetric properties. Examples of applications include classification of patients on the basis of phenotypical prognostic factors, identification of different loads of disease activity to choose therapeutic strategies with the best benefit/harm profiles, identification of patients non-respondent to drugs in order to shift therapy and monitoring the safety of drugs.

Recently, the literature has flourished with an increasing number of studies focused on the search for diagnostic or prognostic biomarkers for MS, also thanks to the availability of proteomics and metabolomic approaches [3], also including narrative reviews and some systematic reviews focused on specific biomarkers. However, a review that systematically summarizes all the findings on this topic is still lacking, but it should take a large bulk of the data on the topic into account.

We aimed to perform a literature review of body fluid biomarkers that have been proposed as prognostic factors for MS. We addressed this aim by performing a scoping review of reviews (umbrella review) on body fluid biomarkers in MS. Although several studies have been performed on this matter, there is a need for an up-to-date summary to put all the knowledge together. Furthermore, we made an evaluation of the reliability of each marker based on power considerations. Reliable biomarkers early predicting the conversion of CIS or RIS to clinically definite MS or the conversion from RRMS to SPMS would be of great help on the choice of MS therapy.

## 2. Methods

Bibliographic search: we searched MEDLINE through PubMed for review articles up to the 25 November 2021. We used two different search strings: (1) “multiple sclerosis progression biomarker *” (1435 papers, 322 of which were review articles); (2) “multiple sclerosis fluid biomarker *” (1581 papers, of which 245 were review articles). We then selected only review articles: 62 reviews were overlapping between the two search strategies, which led us to a total of 505 reviews, published from 1998 to November 2021.

Selection of markers and outcomes: We selected reviews including any type of body fluid biomarker, cell types and tissues. Only reviews assessing biomarkers for prognostic purposes were selected, considering the following outcomes: (1) conversion from CIS or RIS to CDMS (clinically definite MS) or from RIS to CIS; (2) disease severity and progression (defined as any disability outcome: change in EDSS (Expanded Disability Status Scale) or Multiple Sclerosis Severity Scale (MSSS), number of lesions, change in brain volume, any other neurological and/or neuroimaging outcome); (3) disease activity (defined as relapse rate, number of active lesions, or change in biomarker levels between relapse and remitting phases).

Selection of studies: two authors (NB and SD) blindly screened the 505 articles for eligibility by reading the abstracts; disagreements were resolved by discussion. After screening, we selected 154 reviews (spanning from 1998 to November 2021 as of publication date) for further evaluation. As all the selected articles were reviews covering the previous literature, we decided to limit the time range to the last seven years, thus including all papers since 2014 (113 papers). These articles were acquired in full text and blindly evaluated by the same two authors. After evaluation of the full text, 61 papers were excluded because they were deemed not specific to biomarkers in MS or because they only described diagnostic and not prognostic biomarkers. A final number of 52 reviews were thus selected and extensively read for data mining. Original articles were obtained and read to confirm and detail the data reported by the review authors. We reviewed biomarkers supported by: randomized controlled trials; open label extension studies; prospective and retrospective cohort studies; case control studies, including MS patients who progressed and those who did not progress, or measuring the biomarker levels in patients with different prognostic features, and cross-sectional studies measuring the correlation between the level or the concentration of a biomarker and quantitative clinical outcomes (EDSS, MSSS, number of lesions, brain volume…).

Data extraction: we collected the following information for each biomarker: biomarker class, full name and description, body fluid in which the biomarker was measured, prognostic outcome, synthetic description of the observations, number of patients and reference to both the review and original paper(s).

Evaluation of the biomarkers: to evaluate the reliability of each biomarker, we calculated the minimal sample size needed to achieve an 80% power (considering a type-I error of 0.05) of observing at least a 25% increase in the frequency of the examined outcome in patients positive for the biomarker compared to those negative for the biomarker, and we tabulated the list of studies that were based on sample sets with sufficient statistical power. This additional analysis was performed for three outcomes: (1) conversion from RIS to CIS or MS, (2) conversion from CIS to CDMS, (3) conversion from RRMS to SPMS. The baseline frequency of the above outcomes was assessed by referring to literature reports based on large sample sets. The power analysis was performed using MedCalc software version 20.015 (MedCalc Software Ltd., Ostend, Belgium) [4]. Figure 1 shows the flowchart of the selection process.

## 3. Results

The search for review articles yielded 3016 papers, 567 of which were reviews. After the removal of duplicates, 505 review articles remained and were further screened based on inclusion criteria. After screening for eligibility, 52 articles were included in the present review (listed in Table 1).

We identified 110 molecules proposed as prognostic biomarkers, plus seven extracellular vesicle types and 24 immuno-profiles. Biomarker information was grouped into 11 tables (Appendix A) based on arbitrary biomarker classes: “cytoskeleton” (including neurofilaments); “vitamin D”; “chitinase/chitinase-like proteins”; “cytokines, chemokines, TNF-receptor superfamily members, biomarkers of innate immunity”; “antibodies” (including oligoclonal bands, immunoglobulin light chain and viral antibodies); “miRNA”; “oxidative stress biomarkers”; “cholesterol and markers of cholesterol turnover”; “immunoprofile”; “extracellular vesicles” and “other biomarkers”. Of all the observed biomarkers, those reported in Appendix A (cytoskeletal biomarkers) are basically markers of neuronal damage (degenerative biomarkers), while all the others are mainly inflammatory biomarkers (Appendix A). The clinical outcomes were grouped into six main categories, as indicated in the Appendix A in the column “outcome”, namely disease severity (56 entries), disease activity (69 entries), disease progression (15 entries), conversion from RIS to CIS (two entries), conversion from CIS to MS (27 entries) and conversion from RR to SP (five entries). Appendix A also reported the body fluid examined for each biomarker in each study. The majority of the studies were conducted on CSF (75 entries) or on serum (57 entries). There were also studies conducted on plasma (19 entries), whole blood (14 entries), PBMCs (15 entries) or specific blood cell types, such as T-cells. We also examined papers that have considered biomarkers in other body fluids, such as urine or tears, but very few significant results have been reported. For each biomarker, we searched the original research articles cited in the reviews and extracted all the information summarized in the Appendix A, such as the number of patients, length of the follow-up period and statistical data.

## 4. Main Biomarkers

### 4.1. Cytoskeletal Biomarkers (Appendix A)

Neurofilaments: The physiological function of the axon is strictly dependent on the structural layout of the axonal cytoskeleton, which includes a network of interconnected actin microfilaments, neurofilaments and microtubules. Neurofilaments are the most abundant component of the axonal cytoskeleton, and they consist of three subunits that differ in their molecular size: light chain (NfL), intermediate and heavy chain (NfH). Pathological neurodegeneration-causing axonal damage results in the release of neurofilaments in the CSF, thus all three chains can be detected in blood or CSF after neuronal damage. Both NfL and NfH CSF levels are increased in MS patients, however, this is not specific to MS: NfL levels are also increased in other neurodegenerative diseases, such as prion diseases, amyotrophic lateral sclerosis, Alzheimer’s disease, Parkinson’s disease, Hungtington’s disease, and traumatic brain injuries, and they can be influenced by other confounding effects, such as age or non-neurological comorbidities. All these considerations hamper the use of NfL as a diagnostic biomarker [46,56].

Nevertheless, NfLs are currently among the most promising prognostic biomarkers for MS. In particular, the predictive role of both CSF and serum NfL levels for the conversion from RIS or CIS to clinically definite MS has been confirmed by several studies. Moreover, a recent paper detected an increase in the levels of NfL in the serum of patients up to 6 years before the clinical onset of the disease [57]. Some studies also reported an association between NfL levels and disease progression (measured as worsening of EDSS or increase or enlargement in T2 lesions), but the findings seem less consistent [46]. A recent meta-analysis of case-control studies pointed out that NfL levels are also useful in distinguishing patients in an active phase from patients in remission [58].

Serum and CSF NfH levels are elevated both in MS patients and in EAE mice, and higher plasma/serum NfH levels were described in acute optic neuritis with higher levels predicting a worse visual outcome [23]. The evidence that supports NfH as a prognostic biomarker is less convincing than that reported for NfL; however, there are studies that point to this molecule as a marker of bad prognosis and progression to SPMS. Moreover, in a trial of lamotrigine in SPMS, those patients adherent to therapy had significantly lower serum NfH when compared to the non-adherent group, and patients with detectable NfH had a higher EDSS score and performed worse in the 25 foot-walk and 9-hole peg test [23], suggesting a possible use of this marker to monitor lamotrigine therapy.

Other cytoskeletal biomarkers: In addition to neurofilaments, other components of the axonal cytoskeleton or other CNS cytoskeletal proteins have been proposed as biomarkers in MS, namely tau protein and glial fibrillary acidic protein (GFAP).

Tau protein is a microtubule-associated protein located in neuronal axons, and it promotes the assembly and stability of microtubules. It may be involved in the establishment and maintenance of neuronal polarity. Tau protein is accumulated in neurons of patients with neurodegenerative diseases such as Alzheimer’s and Creutzfeldt–Jakob diseases. Increased CSF levels of total or phosphorylated tau protein have been reported in these conditions, reflecting the degree of axonal damage [27]. A 3-year follow-up study reported that RRMS patients with higher CSF Tau levels had a faster progression of disability and reached a higher degree of disability (measured with EDSS) at the end of follow-up [59].

A study that observed increased tau levels in patients with gadolinium-enhancing lesions [60] suggested that CSF tau levels may reflect the inflammation activity occurring during the disease. However, recent studies observed a decrease in tau levels throughout the disease and that SPMS patients, in which brain atrophy is the predominant feature, have lower CSF tau levels compared to RRMS [61]. These observations suggest instead that reduced tau levels may reflect a process of axonal degeneration.

Due to these discrepancies, tau levels are not considered a biomarker for MS at present, and more studies are needed to clarify its possible role as a biomarker of neuronal damage.

GFAP is the major intermediate astrocytic cytoskeletal protein, and it is considered a marker of astrogliosis, a prominent histopathologic feature in MS [27]. A recent meta-analysis confirmed that GFAP levels are higher in MS patients compared to healthy controls and in patients in relapse compared to patients in remission [62]. However, increased GFAP levels have also been detected in patients with NMO or with other neurological conditions, therefore, it is considered a non-specific marker of CNS pathology and its use is probably not suitable for MS diagnosis [12,27]. However, GFAP levels in CSF correlate with measures of disability and with time to reach an EDSS ≥ 3, therefore, this protein has been proposed as a biomarker for disease progression [63].

### 4.2. Vitamin D (Appendix A)

Vitamin D is derived from cholesterol, and it has an immune-modulatory role by controlling the transcription of numerous genes relating to immunity. Low vitamin D intake or low circulating 25-dihydroxyvitamin D (25OH-D) are known risk factors for MS susceptibility. It has also been demonstrated that vitamin D deficiency is also associated with conversion from CIS to SM and that in patients with clinically defined MS, lower vitamin D levels are associated with worse prognosis and with markers of disease activity and progression [64].

### 4.3. Antibodies (Appendix A)

IgG oligoclonal bands (OCB.) CSF-restricted IgG OCB were first reported in MS patients in studies in 1960. The presence of IgG-OCB is not mandatory for MS diagnosis according to the McDonald criteria [65,66]; however, several studies assessed their usefulness in diagnosis. The clinical sensitivity of IgG OCB, as assessed by two meta-analyses, ranges from 88% to 94% [67,68]. In differential diagnosis against other inflammatory diseases of the CNS, IgG-OCB shows a low specificity (61%), which underlies the need for additional biomarkers [15].

The presence of IgG-OCB in CSF is also a known prognostic factor for conversion from CIS to CDMS. A large multicentric study [1] showed a higher risk of conversion from CIS to CDMS in patients with the presence of OCB in CSF and a higher number of lesions in MRI.

CSF-restricted IgG-OCB are robust biomarkers that need no further validation to confirm their role both in diagnosis and in prognosis. However, the lack of an automated quantitative detection method could be a limit for their routine usage in some clinical laboratories [15]. Some authors have proposed that tear OCB detection can replace CSF OCB detection as a less invasive procedure [3]. However, the matter is still controversial [69].

IgM oligoclonal bands. IgM antibodies are involved in the intrathecal B-cell response in MS patients [27]. In most patients with MS, IgM-OCB are directed against myelin lipids. Several studies have observed that the presence of IgM-OCB in CIS patients is predictive of conversion to CDMS and that the main time for conversion is shorter in IgM-OCB positive patients compared to negative patients. However, their added value to IgG OCB as a biomarker is yet to be established [27].

Antibodies against different viruses can be identified in CSF in about 90% of MS patients. In particular, case/control studies evidenced that Ab against components of EBV, measles virus, rubella virus, varicella zoster and human herpes virus 6 (HHV-6) are more often detectable in MS patients than in healthy controls [12,70], thus have a potential value as diagnostic biomarkers. In particular, IgG against the neurotropic viruses (measles, rubella, and varicella zoster, collectively referred to as MRZ) has been reported in the CSF of 80–90% of patients with MS, but only in 5% of patients with NMO, and they are absent in patients with paraneoplastic neurological disorders, thus showing utility in differential diagnosis. Brecht et al. showed an association between the number of positive antibodies and disease duration [71].

Anti-EBNA antibodies and MRZ-specific IgG may also have a prognostic value in predicting conversion from CIS to CDMS (see Appendix A), but this data needs replication in additional cohorts. Moreover, the identification of MRZ-specific IgG is still technically challenging [15], and this may limit their usage.

Synthesis of intrathecal antibodies against JCV (John Cunningham virus) has been reported as a marker of side effects in the therapy with natalizumab [12]. In particular, the risk of developing progressive multifocal leukoencephalopathy increases with the increasing levels of anti-JCV antibodies [70]. However, antibodies against this virus have not been consistently reported as prognostic biomarkers.

FLC (free light chains): Several authors (reviewed by Ramsden [28]) claim that a high kFLC concentration is supportive of MS diagnosis, and this data was confirmed in a recent multicentric study showing a 95% specificity and 95% sensitivity [72]. In diagnosis, the clinical sensitivity and specificity of kFLC are similar to those of the analysis of OCB in CSF that are still considered the “gold standard” [24,28].

A possible problem in the usage of this marker, however, is the high heterogeneity in the techniques used for its quantification, regarding the assay method used (ELISA or nephelometry), the antibody type (polyclonal or monoclonal) and the metric (CSF concentration of kFLC and/or λFLC, FLC quotient: [CSF kFLC]/[serum kFLC] and kFLC index (kappa index): ([CSF kFLC]/[serum kFLC])/([CSF albumin]/[serum albumin] are the three most common methods). Furthermore, the studies also differ in the kFLC cut-offs used.

The Kappa Index is considered an estimate of intrathecal kFLC synthesis, and it is probably a better marker than kFLC concentration because it takes the effect of the blood–brain barrier into account, improving diagnostic accuracy and reducing false positives. Menéndez-Vallarades et al. performed a meta-analysis of seven studies that used the kFLC kappa index (300 MS patients from a total of 1155 samples), observing a global sensitivity of 0.91 (95% CI: 0.88–0.94; *p* = 0.0063), a global specificity of 0.89 (95% CI: 0.87–0.92; *p* < 0.00001) and a global diagnostic Odds Ratio (OR) of 143.65 (95% CI: 39.19–526.50; *p* = 0.0002) [36].

Several studies have proposed the use of CSF and serum FLC as a prognostic biomarker, mainly for the risk of conversion from CIS to MS (Appendix A).

More recently, Kaplan et al. described an association between salivary FLC levels and disease activity in a small case/control study [73]. This data is of interest because of the accessibility of such a body fluid, but more studies are needed to explore the possible use of salivary FLC as prognostic biomarkers [74].

Auto-antibodies: Several autoantibodies have been investigated as diagnostic or prognostic biomarkers in MS. Among these are anti-MOG (Myelin Oligodendrocite Glicoprotein), antibodies for myelin basic protein (MBP) and antibodies directed against neurofilament light chains. Further studies are needed to better elucidate the possible utility of these antibodies as prognostic biomarkers.

Of note, serum antibodies against aquaporin 4 (AQP4) are a strong diagnostic biomarker that allows for distinguishing MS from NMO: anti-AQP4 antibodies are found in 70–80% of NMO patients but not in MS patients [12,15]. No correlations between these antibodies with prognostic outcomes have been reported.

### 4.4. Chitinase and Chitinase 3-like Proteins (Appendix A)

Chitinases are a family of secreted glycoproteins that bind and hydrolyze chitin. Chitinase 3-like-1 (CHI3L1, also called YKL-40) and chitinase 3-like-2 (CHI3L2) are chitin-binding proteins homologous to chitinases but lacking their capacity for chitin hydrolysis. They are expressed in MS brain tissue, particularly in astrocytes in white matter plaques and in normal appearing white matter. CHI3L1 is also expressed in microglia in MS lesions. Chitinase 1 (CHIT1) and CHI3L1 mediate increased immune cell trafficking across the blood–brain barrier, and CHI3L1 is hypothesized to play a role in chronic inflammation and tissue remodeling [26].

Elevated CSF CHI3L1 levels in CIS patients have been validated as a prognostic marker for conversion from CIS or optic neuritis to clinically-defined MS [75,76], and they correlate with a shorter time for conversion from CIS to MS, a more rapid accrual of disability and an increased likelihood of cognitive impairment [26].

The role of CHI3L1 as a biomarker for disease activity or disease progression in clinically definite MS is, instead, less clear. CSF CHIT1 levels are generally higher in MS patients compared to healthy controls, and there are studies that report a correlation between CSF CHI3L1 levels and the rapid development of brain lesions and disability [49]. However, the correlations between CHIT1 and clinical, radiologic and prognostic measures have a lack of consistency between studies, therefore, at present, it is difficult to determine CHIT1’s utility as a biomarker for MS [26].

### 4.5. Cytokines/Chemokines (Appendix A)

The serum and CSF profiles of both cytokines and chemokines are affected by changes related to the disease status or to other factors such as infections and stress, and they are very heterogeneous among patients. Additionally, other inflammatory diseases of the CNS show similar cytokine and chemokine patterns to MS. These observations, until recently, discouraged the clinical routine use of these molecules as diagnostic biomarkers for the early identification of MS. However, in a recent meta-analysis of 226 studies, including 13,526 MS patients and 8428 healthy controls, Bai et al. (showed that 13 CSF and 21 blood cytokines are significantly associated with MS [77]. In particular, CSF CXCL13 levels and blood IL2R and IL-23 levels are consistently different in MS from healthy controls, and they may be employed for diagnostic purposes.

Moreover, CSF levels of cytokines and chemokines could be used to estimate the current status of disease activity and to predict disease progression. The best studied biomarkers belonging to this class are CXCL13 and osteopontin.

CXCL13: Chemokine (C-X-C motif) ligand 13 (CXCL13) is a potent B-cell chemoattractant critical for B-cell migration and for the development of B-cell follicles and secondary lymphoid structures [23]. Increased expression levels in MS patients compared to healthy controls were observed for all disease courses. However, CXCL13 is also increased in NMO and other neuroinflammatory diseases [78], and patients with viral/bacterial infections show extremely high CXCL13 levels [79], therefore, due to low specificity, this chemokine is not suitable as a diagnostic biomarker. CXCL13 has a potential use as a prognostic biomarker instead. A large cohort study demonstrated that increased CXCL13 expression is associated with increased relapse rates, EDSS score and lesion burden [79]. An association of CXCL13 levels with CIS to MS conversion was also described (table E).

Osteopontin: Osteopontin (SPP1 or OPN) is a sialoprotein with pleiotropic roles, including inflammation, T-cell co-stimulation, Th-1 cell polarization and interferon-gamma expression. It is involved in the development and progression of several autoimmune diseases, including MS, and it is expressed in MS lesions [13,23]. OPN levels are increased in MS patients compared to controls [13,23], but this has also been reported for several other neurologic and non-neurologic disorders; therefore, it is not a specific biomarker for MS diagnosis. Several studies report the association of OPN levels with disease activity (table E). Instead, there are controversial results about OPN’s role as a biomarker of disease severity.

### 4.6. miRNA (Appendix A)

MicroRNAs are short (~20 nt), single-stranded, non-coding RNAs which regulate post-transcriptional protein synthesis and immune system function mainly by regulating transcription factors, pro-apoptotic proteins and elements of the signal transduction cascade. MiRNA regulating the development of immune cells shows different levels of expression in the thymus and bone marrow. Dysregulation of miRNAs may play an important role in the mechanism of MS. Many studies have used miRNA screening approaches to identify MS biomarkers, revealing several miRNAs that are up or down regulated in MS patients compared to healthy controls or in different MS clinical subtypes, as a prognostic marker or in response to certain therapies [26,42,80]. For three miRNA (miR-150, miR-181c and miR922), a predictive role for CIS to MS conversion was proposed, but this needs to be validated on a larger sample set.

Other non-coding RNA. Recently, several studies have begun to investigate the role of long noncoding RNAs (lncRNA) and circular RNAs in MS pathogenesis. We are still far from a possible use of these molecules as prognostic biomarkers in MS, however, a lncRNA was found to be dysregulated in patients in remission compared to those in relapse, so these markers may be worth further investigation [81,82].

### 4.7. Cholesterol, Markers of Cholesterol Turnover and Other Lipidic Biomarkers (Appendix A)

Cholesterol is an essential component of cellular and myelin membranes, a cofactor for signaling molecules and a precursor of steroid hormones and vitamin D. Its homeostasis is compartmentalized, with only limited interaction between the brain and blood. Early studies (~1950) have investigated the esterification of cholesterol as a feature of demyelination in white matter and the spinal cord in MS CNS. More recently, markers of cholesterol turnover, such as apolipoproteins and oxysterols, have been investigated for their correlation with several disease severity outcomes [22].

Lipoprotein-bound cholesterol. All studies that measured circulating cholesterol found that elevated circulating LDL and/or total cholesterol are associated with and/or predictive of worsening disease, as assessed by prognostic outcomes including EDSS score, contrast-enhancing lesions, T2 lesion load on MRI and retinal nerve fiber layer thinning [22]. Potential causal mechanisms, however, remain to be elucidated. Furthermore, several authors report that ox-LDL and some components of apolipoprotein particles are more predictive of an adverse outcome than LDL or total cholesterol [22]. In spite of the concordance between studies, its interpretation is still challenged by the fact that all studies were performed on small populations. A bigger effort is needed to investigate the potential utility of these molecules as predictive biomarkers. These potential biomarkers could also be useful in patients’ follow-up because the most significant observations so far are relative to serum/plasma and not to CSF, thus potentially avoiding repetitive lumbar punctures.

Oxysterols. To maintain cholesterol homeostasis, excess cholesterol must be removed from the CNS, enter into blood circulation and be processed by the liver. The first step in cholesterol metabolism is the oxidation and subsequent conversion to 27-hydroxycholesterol (27-OHC) by the enzyme CYP27A1 outside the CNS and to 24-hydroxycholesterol (24-OHC) in neuronal and glial cells. 24-OHC can cross the blood-brain barrier (BBB). Because most brain cholesterol is contained within the myelin, elevated levels of circulating or CSF 24-OHC likely reflect changes in brain cholesterol turnover caused by demyelination [22].

Apolipopreoteins. ApoE is a minor component of HDL, and it can act to remove cholesterol from injured nerves and to promote axonal regeneration and remyelination, and to reduce oxidation and inflammation [22]. However, there is no clear evidence for an association between ApoE levels and disease activity, and results have been equivocal so far.

Sphingolipids. Sphingolipids, as components of the lipid bilayer, are particularly abundant in the CNS and are involved in all the processes of exo and endocytosis and cell signaling, therefore, the regulation of their network is crucial for the proper function of the CNS. Alterations in the sphingolipid pathway may reflect disease activity, in particular, oligodendrocyte damage and acute demyelination [53]. A number of ex-vivo studies, (reviewed by Podbielska, 2021 [53]) have investigated the role of lipid signature in the CNS as a potential biomarker for disease activity and disease progression, by comparing lipidomic profiles in demyelinating lesions with normal-appearing CNS or CNS normal-appearing tissue in patients with active disease compared to inactive MS or healthy individuals. Moreover, similar studies observed altered levels of glycerolipids in the plasma or CSF of MS patients compared to healthy controls or other neurological diseases. However, there is still little evidence of molecules that can be of utility in the follow-up of MS patients.

### 4.8. Oxidative Stress Biomarkers (Appendix A)

Oxidative damage to DNA, protein and lipids is a major feature of MS neuropathology in both relapsing-remitting and progressive disease. Reactive oxygen species, the mediators of oxidative damage, are released as part of the respiratory burst of activated neutrophils, monocytes and microglia and result in oligodendrocyte injury and axonal damage [19].

Recently, a systematic review on oxidative stress molecules as MS biomarkers was performed [19]. The proposed biomarkers for MS prognosis were nitric oxide, superoxide dismutase, and several catalases and products of peroxidation. However, all the studies performed on this topic could avail of small population, therefore, further analysis on bigger sample sets is needed.

A bigger study observed a correlation between EDSS and lactate, a product of anaerobic metabolism, however, the correlation coefficient was poor (table H).

### 4.9. Immunoprofile (Appendix A)

Flow cytometry studies of CSF have helped to elucidate the pathophysiological mechanisms of MS. It is generally accepted that the migration of leukocytes, specifically memory T cells, into the CNS is a crucial step in the disease. Alvermann et al. presented a review of the cellular alterations in the CSF and peripheral blood in MS patients [5]. Most alterations have a value for diagnosis or differential diagnosis, as they differentiate MS patients from healthy controls or patients with other neurological conditions, or they differentiate different clinical MS forms. Moreover, there are also differences in immunophenotype that correlate with disease activity. Of interest, the expression of CD5 on B cells in peripheral blood is predictive of conversion from CIS to MS [5].

### 4.10. Extracellular Vesicles (Appendix A)

Extracellular vesicles (EVs) are small, non-nucleated vesicles derived from various cell types, and they usually contain several biomolecules such as proteins and miRNAs. Studies conducted in the past two decades confirm that EVs play a role in various human diseases, including inflammatory diseases and, in particular, MS [47]. Several studies have evaluated the role of EVs as diagnostic or prognostic biomarkers for MS, focusing on their content and surface markers [50]. Some of these studies demonstrated a good correlation with disease activity or disability parameters. However, none of the studies conducted so far has reached the target of a possible use of the vesicles in clinical practice [51].

### 4.11. Other Biomarkers (Appendix A)

Matrix metalloproteinase-9 is a member of the zinc metalloproteinase protein family (MMPs). MMPs control cell migration across the blood–brain barrier by disrupting the subendothelial basement membrane and other components of the extracellular matrix and eventually affect myelin destruction and axonal damage in MS [23,27]. Increased expression of various MMPs (MMP−2, −3, −7 and −9) has been demonstrated in autopsied MS brains, and MMP9 was detected in acute MS lesions [83].

Several studies investigated the relationship of MMPs or their tissue inhibitors (TIMPs) with disease activity by comparing patients that showed active disease at a certain time with stable patients.

Neurotrophins can stimulate neuronal regeneration and repair. BDNF is expressed in the areas of the brain involved in learning and memory [84] but also in immune cells [23], and it was found in MS active lesions in infiltrating T cells and macrophages and circulating lymphocytes as well as in neurons and activated astrocytes [85,86]. Some studies reported an association with disease activity outcomes, but these findings need more investigation.

Complement components. The complement system has an established role in the pathogenesis of MS, as evidenced by the deposition of complement components and activation products in the white matter plaques in brain tissue. The presence of complement, antibodies and Fc receptors in phagocytic macrophages suggests that complement mediated myelin phagocytosis is the dominant mechanism of demyelination in clinically definite MS [29].

Several studies have tried to identify an association of complement components, both from classical and alternative pathways, with disease activity and clinical phenotypes. A promising result was observed for factor H, a major regulator of the alternative pathway. Serum factor H levels could be of utility as a diagnostic biomarker to distinguish SPMS from RRMS, with a sensitivity of 89.41%, a specificity of 69.47% and a positive predictive value of 72.38% [29]. Moreover, factor H levels increase progressively with disease progression in patients that are transitioning from relapsing to progressive disease [87].

Fetuin A: Fetuin A is a serum glycoprotein synthesized by hepatocytes. It is involved in endocytosis, brain development, formation of bone tissue, calcium metabolism opsonization and immune regulatory functions. Several studies proposed this protein as a biomarker for differential diagnosis, prognosis and treatment response [12]. However, the findings reported so far are discordant: higher levels of fetuin-A are found during acute inflammatory episodes and in demyelinated lesions, but lower levels in CIS patients are associated with an earlier conversion to CDMS. Furthermore, it has been reported that RRMS patients have lower CSF fetuin-A levels than healthy controls, while for SPMS patients, it is the opposite [12]. Further studies, with a higher number of patients, should be performed to clarify the possible role of this molecule as a biomarker of MS.

14-3-3 protein: The 14-3-3 protein family comprises several phosphoserine-binding proteins expressed in neurons and glial cells. Its subunits are involved in apoptosis, cell cycle control and neuronal development [27]. This protein has been proposed as a prognostic marker for a more rapid conversion from CIS to SM. However, the same authors conclude that, despite 14-3-3 protein seeming to be a specific indicator of lower conversion time and poor prognosis, its detection had very low sensitivity. Therefore, the authors did not consider it reasonable to perform a lumbar puncture only for 14-3-3 determination [88].

Adhesion molecules: The adhesion of cells is important for tissue formation and the infiltration of tissue by immune cells. Proteins of the cell adhesion molecules family (CAMs) exist as membrane-bound and soluble forms and are involved in cell–cell contact. An impairment of the expression of these proteins can increase the permeability of the blood–brain barrier and promote migration of immune cells into the CNS. It has been observed that CSF levels of VCAM and ICAM correlate with disease activity [12].

Neuronal cell adhesion molecules (NCAM) are involved in axonal outgrowth, guidance and fasciculation. They have been proposed as a diagnostic biomarker for MS, as CSF levels are higher in MS patients compared to controls [6]. Furthermore, a study observed decreasing levels of NCAM in the CSF in different MS subtypes following a stepwise manner in the order (CIS > RRMS > SPMS) [27]. In line with this observation, NCAM levels are inversely correlated with disease severity, assessed with an EDSS score [89].

N-acetylaspartate (NAA) is an amino acid expressed specifically in neurons, neuronal processes and oligodendrocytes. Due to its tissue specificity it has been used as a biomarker of axonal damage. CSF NAA levels are decreased in SPMS compared to RRMS and CIS, suggesting that NAA levels diminish during late phases of the disease. A correlation with disability and MRI outcomes has been observed, and NAA levels are lower in patients with higher EDSS, lower brain volume and increased lesion load. However, the high variability between different MS subtypes probably makes this molecule unsuitable to be used as a single and independent biomarker [27]. In this line, some research groups have proposed the combination of NAA with other biomarkers of axonal damage to evaluate different phases of neuronal damage occurring throughout the disease course (for example, Teunissen et al. tested NAA in combination with NfL in 2009) [15].

Glutamate is the major excitatory neurotransmitter of the CNS. Glutamate signaling regulates intracellular ionic homeostasis and it has an important role in the communication network established between neurons, astrocytes, oligodendrocytes and microglia. Impairments in glutamate homeostasis can affect many functions and interactions in CNS cells and lead to excitotoxicity, one of the mechanisms involved in MS pathogenesis [27]. A study on a small population reported that, in SPMS patients, glutamate levels are higher in patients who showed progression of neurologic disability in the last 6 months of follow-up [90]. If confirmed, this molecule could be an interesting biomarker because there is little evidence of biomarkers specific for disease progression in SPMS disease. However, further studies are needed to evaluate the usefulness of glutamate as a prognostic biomarker in MS.

## 5. Study Power Evaluation of Biomarkers Studies

Many studies on prognostic biomarkers have been performed on small sample sets. Therefore, in order to identify reliable prognostic biomarkers, we calculated the minimum sample size needed to achieve an 80% power of observing at least a 25% increase of the frequency in each outcome in patients positive for the biomarker compared to those negative for the biomarker.

This analysis was performed for the following three outcomes, considering different estimates of the event rate:(1)Conversion from RIS to CIS or MS: considering a frequency of conversion of 28% based on Kantarci et al. (2016) [2], who followed 453 RIS subjects for 5 years, we calculated a minimal sample size of 1382.(2)conversion from CIS to CDMS: considering a frequency of conversion of 59.5% based on Khule et al. (2015) [1], who followed 1047 CIS for 4 years, we calculated a minimal sample size of 330.(3)conversion from RRMS to SPMS: the estimated rate of conversion is 15.4% at 10 years based on Barzegar et al. [91], who followed 1903 RRMS, and 66.3% at 28 years based on Scalfari et al. (2011) [92], who followed 806 RRMS for a mean time of 28 years. Therefore, we estimated a minimal sample size of 2958 and 214, respectively, for follow-up periods of 10 years and approximately 25–30 years.

We observed only six studies with adequate power, and all dealt with the rate of conversion from CIS to CDMS (Table 2). Furthermore, a study analyzed the association between complement factor H levels and conversion from RR to SP on 350 MS subjects, which would have been adequate for long follow-up periods, but the follow-up time was too short (2 years) [87].

## 6. Conclusions

Due to the need to find a reliable body fluid biomarker for predicting prognosis in MS, many molecules have been investigated, and this makes the task of performing a comprehensive review of these biomarkers very challenging. To our best knowledge, this is the first scoping review on this topic.

Since most studies were conducted on small sample sets and they could achieve only nominal significance levels, the results currently available for many of these molecules are still scarcely reliable. Another problem is the high heterogeneity among the different studies in the quantification of the biomarker, the statistical analyses used, the length of the follow-up and the assessment of the outcome (clinical feature or surrogate outcome, such as MRI). This heterogeneity often did not allow for the performance of a meta-analysis of different studies that could provide more consistent data about the effective predictive value of a biomarker. This observation highlights the need for common guidelines that should be applied when performing or reporting a study on MS prognostic biomarkers.

Currently, the most promising prognostic biomarkers for the prediction of conversion from CIS to clinically definite MS are CSF and serum levels of NfL (due to the large number of confirming studies), CSF levels of IgG-OCB, the kFLC kappa index and CHI3L1, and serum levels of vitamin D. Conversely, there are very few molecules that have been proposed to assess the disability accrual over time and to predict the conversion from RRMS to SPMS, thus this clinical step is still very tricky, and it still needs exploration. As a matter of fact, disease prognosis can be assessed using clinical measures including the number of relapses and a certain score on the EDSS scale. Although EDSS is widely used, it has well-recognized limitations and weaknesses in reliability and sensitivity to change [93]. Another possible way of measuring prognosis is to estimate the time of transition from an RR phase into an SP phase, but there is a wide range of literature showing how difficult it is to estimate the precise time of transition. In this regard, a recent paper performed a systematic review of the literature and concluded that, at present, no neurophysiological or fluid biomarkers are sufficiently validated to support the early diagnosis of SPMS but that a combination of neurophysiological and fluid biomarkers may be more sensitive in detecting SPMS conversion [45]. Indeed, since this is one of main unmet clinical needs in MS, because biomarkers can identify the optimal time window to interfere, further investigations in this field is strongly advisable.

Among the prognostic biomarkers, special attention should be paid to the molecules that provided reliable results when measured in fluids different from the CSF, to avoid repetitive lumbar puncture in the patients. At the moment, the most promising of such biomarkers is vitamin-D, whose serum levels correlate with disease activity and predict CIS to MS conversion and serum levels of neurofilaments. However, other biomarkers, such as markers of cholesterol turnover, may be worth further investigation, and technological improvements should help in the near future.

Finally, as the studies selected for this review were other reviews, there are probably many other molecules that have been evaluated for their possible usage as prognostic biomarkers in MS that have never been reviewed before and, therefore, they are not included in the present review. Among these, there are probably genetic biomarkers. Additional open issues in the field of biomarkers for MS prognosis can be mentioned. As an example, for many biomarkers, a precise pathognomonic cut-off point for MS has not yet been defined. Moreover, disease-modifying therapies may affect the levels of selected biomarkers. Therefore, it would be useful to identify specific biomarkers at the onset of the disease in the cohort of treatment-naïve patients.

## Figures and Tables

**Figure 1 jpm-12-01430-f001:**
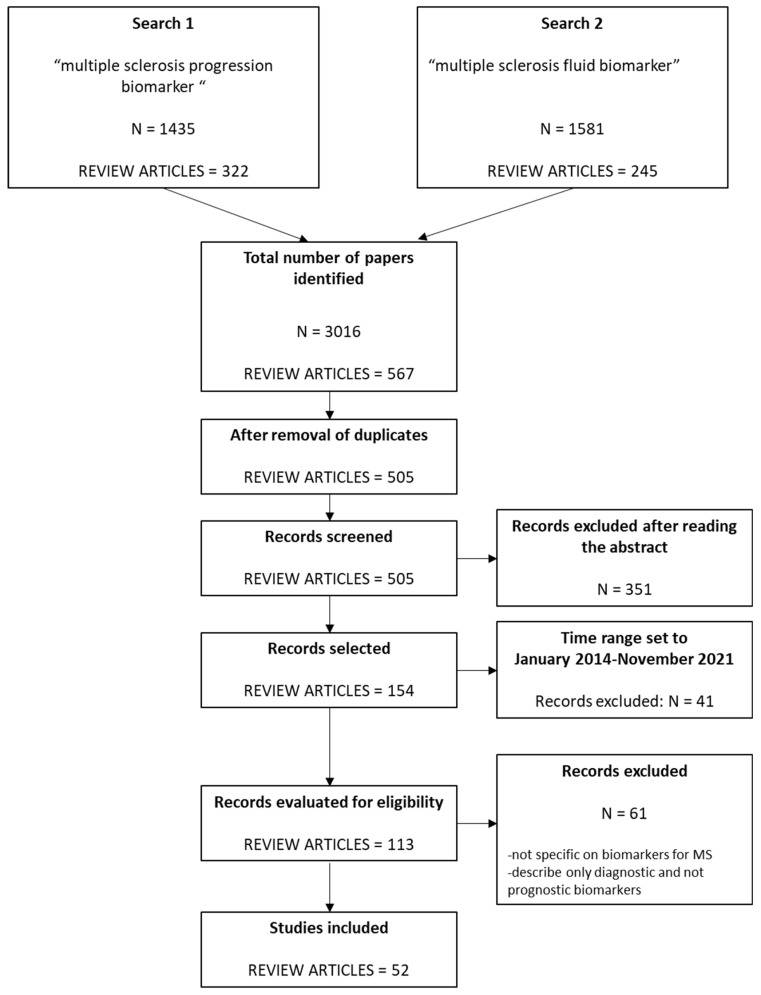
Flowchart of the article selection.

**Table 1 jpm-12-01430-t001:** Review articles included in the scoping review and main topics covered by each review.

Study	Article Type	Main Topic
Alvermann 2014 [5]	Comprehensive narrative review	Cellular alterations in CSF and peripheral blood in MS
Comabella 2014 [6]	Narrative review	Generic, about body fluid biomarkers in MS
Gnanapavan 2014 [7]	Narrative review	Generic, about body fluid biomarkers in MS
Karussis 2014 [8]	Narrative review	Generic, more focused about MS diagnostic criteria
Tomioka 2014 [9]	Narrative review	Generic, about body fluid biomarkers in MS
Abdelhak 2015 [10]	Narrative review	Cytoskeletal damage in MS and Amiotrophic Lateral Sclerosis
D’Ambrosio 2015 [11]	Narrative review	Generic, about peripheral blood biomarkers in MS
Fitzner 2015 [12]	Narrative review	Generic, about CSF biomarkers in MS
Housley 2015 [13]	Narrative review	Generic, about biomarkers in MS
Raphael 2015 [14]	Narrative review	Generic, about body fluid biomarkers in MS
Teunissen 2015 [15]	Narrative review	Generic, about body fluid biomarkers in MS
Axisa 2016 [16]	Narrative review	Generic, on biomarkers and treatment in MS
Comabella 2016 [17]	Narrative review	Generic, more focused on biomarkers for treatment
Häggmark 2016 [18]	Narrative review	About proteomic studies in 5 neurological diseases including MS
Ibitoye 2016 [19]	Narrative review	Oxidative stress biomarkers in MS
Stoicea 2016 [20]	Narrative review	miRNA as biomarkers in neurological diseases
Vermersch 2016 [21]	Narrative review	Generic, about both MRI and molecular biomarkers in MS
Zhornitsky 2016 [22]	Comprehensive narrative review	Cholesterol and cholesterol turnover biomarkers in MS
Barro 2017 [23]	Narrative review	Generic, about both electrophysiological outcomes and molecular biomarkers in MS
Basile 2017 [24]	Narrative review	Free light chain as biomarkers in several diseases including MS
El Ayoubi 2017 [25]	Narrative review	Generic, about body fluid biomarkers in MS
Harris 2017 [26]	Narrative review	Generic, about body fluid biomarkers in MS
Matute-Blanch 2017 [27]	Narrative review	Generic, about body fluid biomarkers in several neurological conditions, including MS
Ramsden 2017 [28]	Narrative review	Free Light Chains as biomarkers in MS
Tatomir 2017 [29]	Narrative review	Complement system as biomarker of disease activity and response to treatment in MS
Thouvenot 2018 [30]	Narrative review	Generic, about body fluid biomarkers in MS
Deisenhammer 2019 [31]	Narrative review	Generic, about CSF biomarkers in MS
Dolei 2019 [32]	Narrative review	Human endogenous retroviruses as possible biomarkers in neurodegenerative diseases
Domingues 2019 [33]	Narrative review	Neurofilament Light chains as biomarker in MS
Gaetani 2019 [34]	Narrative review	Neurofilament Light chains as biomarker in several neurological diseases, including MS
Gudowska-Sawczuk 2019 [35]	Narrative review	Free Light Chains as diagnostic biomarker in MS and HIV infection
Menéndez-Valladares 2019 [36]	Systematic review with meta-analysis	IgG K-index as biomarker in MS
Piket 2019 [37]	Comprehensive narrative review	Small non-coding RNA as biomarkers in MS
Singh 2019 [38]	Narrative review	Biomarkers discovered trough proteomic approaches
Smolders 2019 [39]	Narrative review	Vitamin D and disease activity in MS
Varhaug 2019 [40]	Narrative review	Neurofilament Light chains as biomarker in MS
Ziemssen 2019 [41]	Narrative review	Generic, about body fluid biomarkers in MS
Martinez 2020 [42]	Narrative review	miRNA in blood and CSF as biomarkers for MS
Arneth 2021 [43]	Review article. The search was performed with a systematic method, but reports narrative results only for a selected number of studies (16), including other reviews	Generic, including MRI and molecular biomarkers in MS.
Barboza 2021 [44]	Narrative review	Radiological Isolated Syndrome (including markers of conversion to MS)
Ferrazzano 2021 [45]	Systematic review	Fluid biomarkers for SPMS diagnosis
Ferreira-Atuesta 2021 [46]	Narrative review	Neurofilament Light chains in MS
Gutiérrez-Fernández 2021 [47]	Narrative review	Extracellular vesicles as biomarkers in MS
Jafari 2021 [3]	Narrative review	Biomarkers discovered trough proteomic and metabolomic approaches
Kouchaki 2021 [48]	Narrative review	Neurofilament Light chains mainly as diagnostic biomarker in MS
Mathur 2021 [49]	Narrative review	Generic, about both MRI and molecular biomarkers in MS
Manu 2021 [50]	Narrative review	Extracellular vesicles as biomarkers in MS
Marostica 2021 [51]	Narrative review	Extracellular vesicles as biomarkers in MS
Pietrasik 2021 [52]	Narrative review	miRNA as biomarkers to distinguish RRMS from SPMS
Podbielska 2021 [53]	Narrative review	Lipid biomarkers in MS
Pukoli 2021 [54]	Narrative review	Kineurines and Neurofilament Light chains in MS
Sandi 2021 [55]	Narrative review	Kineurines as biomarkers in MS and other neurological diseases

**Table 2 jpm-12-01430-t002:** Studies with adequate power.

Outcome	Biomarker	Primary Study	Sample Size
CIS to MS	25-hydroxy (OH) vitamin D	Kuhle 2015 [1]	1047 CIS
CIS to MS	25-hydroxy (OH) vitamin D	Ascherio 2014 [64]	468 CIS
CIS to MS	IgG OCB	Kuhle 2015 [1]	1047 CIS
CIS to MS	IgG OCB	Tintore 2008 [39]	415 CIS
CIS to MS	k FLC—K index	Menendez-Valladares 2019 [36]	334 CIS
CIS to MS	CHI3L1	Cantò 2015 [75]	813 CIS

## Data Availability

No new data were created or analyzed in this study. Data sharing is not applicable to this article.

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
