# Peer review of "A Scoping Review on Body Fluid Biomarkers for Prognosis and Disease Activity in Patients with Multiple Sclerosis"

_jpm, 2022, doi:10.3390/jpm12091430_

Round 1

Reviewer 1 Report

This is an article of very high interest and importance .  The lack of fluid biomarkers in MS  makes diagnosis, differential diagnosis and prognosis a real challenge . Finding fluid biomarkers, especially for the prognosis of CIS to MS conversion (diagnosis) or conversion for RRMS to progressive MS could help in better management of the patients and treatment decisions. Finding ‘’fluid’’ biomarkers and even more, serum biomarkers that can be easily measured would be ideal.

From the other side, MS pathology is so complicated, that many different markers can be candidates which of course raises the number of them but also may lead to ‘’over estimation’’ of many of them. Their role and its importance in MS pathophysiology should be taken in consideration before creating long lists. 

In this article authors try to cover relatively big number of those biomarkers.  For some of those markers, their role in MS pathogenesis is minimal or they are known just to be ‘’involved’’ and is still difficult to make a direct correlation with them, their levels and MS prognosis i.e. lipid biomarkers. Others are known and clearly related to MS pathogenesis and activity but cannot easily be related to the disease prognosis. A large number of biomarkers and logically altered during MS , as an epiphenomenon of other changes and they do not reflect a primary MS activity marker (i.e. glutamate, oxitative biomarkers). 

My suggestion is that authors should choose biomarkers that are more important and analyze them in more details and could add a shorter list with other biomarkers that could be related or candidates but without strong correlation. 

Similarly – it would be useful to make a categorization of those biomarkers: those related to disease activity, those related to progression (i.e. inflammatory, degenerative).

The final results of the authors, mention the studies and biomarkers, regarding CIS to MS. Although in the text, relation with ‘’prognosis’’ is mention in many of them, is not analyzed at the end. Diagnosing MS and the possibility of CIS conversion to MS is of course important but most of MS specialists feel that have enough tools and skills for that. The main unmet need in this filed is the diagnosis of progression, the conversion of relapsing form to progressive form and even more the transition from one to the other and the time window to interfere. In this case, relation of early diagnosis of Progressive MS should be added in results and conclusions.

Finally- at the end, authors choose those markers that are related to CIS conversion to MS. In the abstract NFL’s are not mentioned and the n conclusions are mentioned ‘’due to the large number of confirming studies’. Is not clear how these biomarkers were chosen: if it was only based on the number of the patients? Specificity and sensitivity of each should be taken in account. Is also problematic that ‘’number of patients ‘’ was based on the numbers mentioned in other reviews. Is this the reason only few references are mentioned for each biomarker ?Should also be useful to mention in the text the number of patients included in each study ( fe in another study…by … which included X… patients )

Author Response

Dear Reviewer

please find enclosed the revised version of our manuscript entitled “A scoping review on body fluid biomarkers for prognosis and disease activity in patients with Multiple Sclerosis ” in which we have addressed your comments

We will also upload one copy of our text showing track changes and one clean copy

The point by point reply to your comments is reported below

We hope that our revised version has taken into account your comments in a satisfactory manner and that these changes have improved the quality of the manuscript to be suitable for publication in JPM .     

Best regards,

On behalf of all Authors

Nadia Barizzone

Department of Health Sciences, Center on Autoimmune and Allergic Diseases (CAAD), UPO, University of Eastern Piedmont, A. Avogadro University, Novara, Italy, Via Solaroli 17, 28100 Novara; Italy. E-mail: [email protected]

REVIEWER 1

C. This is an article of very high interest and importance .  The lack of fluid biomarkers in MS  makes diagnosis, differential diagnosis and prognosis a real challenge . Finding fluid biomarkers, especially for the prognosis of CIS to MS conversion (diagnosis) or conversion for RRMS to progressive MS could help in better management of the patients and treatment decisions. Finding ‘’fluid’’ biomarkers and even more, serum biomarkers that can be easily measured would be ideal.

R We thank the reviewer for these encouraging comments

C. From the other side, MS pathology is so complicated, that many different markers can be candidates which of course raises the number of them but also may lead to ‘’over estimation’’ of many of them. Their role and its importance in MS pathophysiology should be taken in consideration before creating long lists. 

In this article authors try to cover relatively big number of those biomarkers.  For some of those markers, their role in MS pathogenesis is minimal or they are known just to be ‘’involved’’ and is still difficult to make a direct correlation with them, their levels and MS prognosis i.e. lipid biomarkers. Others are known and clearly related to MS pathogenesis and activity but cannot easily be related to the disease prognosis. A large number of biomarkers and logically altered during MS , as an epiphenomenon of other changes and they do not reflect a primary MS activity marker (i.e. glutamate, oxitative biomarkers). 

My suggestion is that authors should choose biomarkers that are more important and analyze them in more details and could add a shorter list with other biomarkers that could be related or candidates but without strong correlation. 

R We thank the reviewer for these comments and suggestions.

As reported in the introduction, the main aim of this study was to produce a comprehensive picture of biomarkers studied for disease prognosis in MS, without any a-priori selection.  Our goal is to provide a tool for the scientific community useful to evaluate the most investigated and promising markers worth of further follow-up, as well as to ascertain relevant clinical outcomes still lacking adequate biomarkers. We addressed this aim performing a scoping review of reviews (umbrella review). Although several studies have been performed on this matter, including narrative or systematic reviews on specific markers, there is the need of an up to-date summary of all the knowledge in this field. To our best knowledge, this is the first scoping review on this topic.

In our view, the categorization of the “importance” of biomarkers should be based on solid statistical basis. Since most of the studies have been performed on small sample sets, there is a clear risk they are underpowered. For this reason, we calculated a minimal sample size as described at paragraph 5. We were able to perform this extra analysis for three outcomes: RIS to CIS, CIS to SM and RR to SP (table 2) because we know a precise estimation of the conversion rate, measured on literature big datasets. For these three outcomes we observed 6 studies with adequate power, and interestingly all of this were related to the CIS to SM conversion. Although others may add in the near future, the biomarkers listed in table 2 are those that in our opinion carry a higher potential to be used in clinical practice.

C. Similarly – it would be useful to make a categorization of those biomarkers: those related to disease activity, those related to progression (i.e. inflammatory, degenerative).

R. We have added more information in the text related to the supplementary tables, were all the information requested by the referee are described. In details, we did perform a categorization of each outcome, that has been clearly reported in the Supplementary tables 1-11. In details, the clinical outcomes were grouped in 6 main categories, as indicated in the supplementary tables in the columns “outcome”: disease severity (56 entries), disease activity (69 entries), disease progression (15 entries), conversion from RIS to CIS (2 entries), conversion from CIS to MS (27 entries), conversion from RR to SP (5 entries). We added this short description also in the main text in the results section, to guide the reader. The biomarkers have been categorized basing on the molecular type in 11 categories (grouped in the Supplementary tables 1-11 as explained in the main text). In brief, the biomarkers reported in the Supplementary table 1 (cytoskeletal biomarkers) are basically markers of a neuronal damage (degenerative biomarkers), while all the other are mainly inflammatory biomarkers. We added this sentence in the main text and commented in the discussion.

C. The final results of the authors, mention the studies and biomarkers, regarding CIS to MS. Although in the text, relation with ‘’prognosis’’ is mention in many of them, is not analyzed at the end. Diagnosing MS and the possibility of CIS conversion to MS is of course important but most of MS specialists feel that have enough tools and skills for that. The main unmet need in this filed is the diagnosis of progression, the conversion of relapsing form to progressive form and even more the transition from one to the other and the time window to interfere. In this case, relation of early diagnosis of Progressive MS should be added in results and conclusions.

R. We conducted our scoping review in a semi-systematic way, which means that we obtained a picture of all the studies that have been performed, yielding significant results, on this matter. We also report results for molecules proposed as biomarkers for RR to SP conversion. What our study pinpointed is that there are still few studies on biomarkers for this clinical outcome, and that many of these were underpowered due to the small number of patients or to the short follow-up period.

We further stressed this concept adding the sentence in the conclusion “ Indeed, since this is one of main unmet clinical needs in MS because biomarkers can identify the optimal time window to interfere, further investigations in this field is strongly advisable”

C. Finally- at the end, authors choose those markers that are related to CIS conversion to MS. In the abstract NFL’s are not mentioned and the n conclusions are mentioned ‘’due to the large number of confirming studies’. Is not clear how these biomarkers were chosen: if it was only based on the number of the patients? Specificity and sensitivity of each should be taken in account. Is also problematic that ‘’number of patients ‘’ was based on the numbers mentioned in other reviews. Is this the reason only few references are mentioned for each biomarker ?Should also be useful to mention in the text the number of patients included in each study ( fe in another study…by … which included X… patients ).

R. Actually, the reported data (such as the number of patients, length of the follow-up period and the significance value and/or effect sizes) were not based only on the review articles, but they have been obtained and extracted by searching and reading all the primary research articles cited in the reviews. The reviews have been utilized only as a first-pass search tool. The complete information regarding each biomarker is reported in the Supplementary tables 1-11, and only a brief summary has been reported in the main text. The complete list of references mentioned for each biomarker, including all the primary studies used for the review, is also listed in the Supplementary tables and at the end of the Refences section (only the main papers have been cited also in the main text), and it includes a total of 278 original and review papers. The Supplementary tables contain information for all the biomarkers that have been described in the original primary research articles cited in the reviews, and no further selection was performed to examine articles or display results in the Tables. We clarified these aspects in the main text.

We noticed that most of the studies have been performed on small sample sets, clearly underpowered for this kind of studies. Therefore, at the end of the paper we performed a power analysis to classify all the studies on the basis of the study power. As written above, we were able to perform this extra analysis for three outcomes: RIS to CIS, CIS to SM and RR to SP (table 2). This is not an arbitrary choice, but it is due to the fact that for these three outcomes we know a precise estimation of the conversion rate, measured on literature big datasets. The other outcome categories (progression, severity, and activity) are actually wide and diverse categories including many different outcomes, that are often described, measured and studied in different ways in each study, and this makes a systematic analysis almost impossible. For the three outcomes considered for this extra analysis (RIS to CIS, CIS to SM and RR to SP) we observed 6 studies with adequate power, and interestingly all of this were related to the CIS to SM conversion. Unfortunately, there are no significant studies with adequate sample size or follow-up period for the RR to SP outcome, which was one of our main interests since it is one of the main unmet clinical needs in MS. As cited in the main text, a study analyzed the association between complement factor H levels and conversion from RR to SP on 350 MS subjects, which would have been adequate for long follow-up periods (10 years), but the follow-up time was too short (2 years).

Many studies on neurofilaments have been reported, consistently proposing the predictive role of both CSF and serum NfL levels for the conversion from RIS or CIS to clinically definite MS and an association between NfL levels and disease progression. This is the reason why we mentioned these promising biomarkers in the Summary and in the conclusion. However, since none of these studies was sufficiently powered, we were not able to list these biomarkers in the table of powered studies (table 2)

Reviewer 2 Report

This review briefly presents the current state of knowledge in relation to potential multiple sclerosis biomarkers. It is well written, although it does not bring a new perspective on the current state of knowledge in this subject. Furthermore, the following issues require clarification:

1.       There are no references for the first sentences in the introduction describing general information about MS.

2.       There is no mention that many biomarkers have so far not defined a precise pathognomonic cut-off point for MS. Moreover, disease-modifying therapies may affect the levels of selected biomarkers, e.g. neurofilaments, which can be considered also as a biomarker of treatment. Therefore, it would be useful to identify specific biomarkers at the onset of the disease in the cohort of treatment-naïve patients.

3.       For some molecules, e.g. glutamate, it is not stated whether serum or CSF levels are considered in the context of the possible MS biomarker.

Author Response

Dear Reviewer

please find enclosed the revised version of our manuscript entitled “A scoping review on body fluid biomarkers for prognosis and disease activity in patients with Multiple Sclerosis ” in which we have addressed your comments

We will also upload one copy of our text showing track changes and one clean copy

The point by point reply to your comments is reported below

We hope that our revised version has taken into account your comments in a satisfactory manner and that these changes have improved the quality of the manuscript to be suitable for publication in JPM .     

Best regards,

On behalf of all Authors

Nadia Barizzone

Department of Health Sciences, Center on Autoimmune and Allergic Diseases (CAAD), UPO, University of Eastern Piedmont, A. Avogadro University, Novara, Italy, Via Solaroli 17, 28100 Novara; Italy. E-mail: [email protected]

REVIEWER 2

This review briefly presents the current state of knowledge in relation to potential multiple sclerosis biomarkers. It is well written, although it does not bring a new perspective on the current state of knowledge in this subject. Furthermore, the following issues require clarification:

  1. There are no references for the first sentences in the introduction describing general information about MS.

We added  a general reference (https://www.nationalmssociety.org/ )

  1. C.There is no mention that many biomarkers have so far not defined a precise pathognomonic cut-off point for MS. Moreover, disease-modifying therapies may affect the levels of selected biomarkers, e.g. neurofilaments, which can be considered also as a biomarker of treatment. Therefore, it would be useful to identify specific biomarkers at the onset of the disease in the cohort of treatment-naïve patients.

R. We thank the reviever for these considerations that we added in the following sentence in the conclusion “Additional open issues in the field of biomarkers for MS prognosis can be mentioned. As an example, for many biomarkers a precise pathognomonic cut-off point for MS has not yet defined. Moreover, disease-modifying therapies may affect the levels of selected biomarkers. Therefore, it would be useful to identify specific biomarkers at the onset of the disease in the cohort of treatment-naïve patients”

  1. C. For some molecules, e.g. glutamate, it is not stated whether serum or CSF levels are considered in the context of the possible MS biomarker.

R The body fluid type for all biomarkers have been reported in the Supplementary Tables. The majority of the studies were conducted on CSF (75 entries reported in the supplementary tables) or on serum (57 entries). There are also studies conducted on plasma (19), whole blood (14), PBMCs (15) or reporting markers measured on specific blood cell types, such as T-cells. We examined also papers that have considered biomarkers in other body fluids, such as urine or tears, but very few significant results have been reported. We also added a sentence in the main text that summarizes these observation.